# *PRAMEY*: A Bovid-Specific Y-Chromosome Multicopy Gene Is Highly Related to Postnatal Testicular Growth in Hu Sheep

**DOI:** 10.3390/ani12182380

**Published:** 2022-09-12

**Authors:** Shengwei Pei, Fang Qin, Li Wang, Wanhong Li, Fadi Li, Xiangpeng Yue

**Affiliations:** 1Key Laboratory of Grassland Livestock Industry Innovation, Ministry of Agriculture and Rural Affairs, Engineering Research Center of Grassland Industry, Ministry of Education, State Key Laboratory of Grassland Agro-Ecosystems, College of Pastoral Agriculture Science and Technology, Lanzhou University, Lanzhou 730020, China; 2School of Pharmacy, Lanzhou University, Lanzhou 730000, China

**Keywords:** *PRAMEY*, CNV, testicular size, Y-chromosome, sheep

## Abstract

**Simple Summary:**

*PRAMEY* (preferentially expressed antigen in melanoma, Y-linked) is a cancer-testis antigen and is believed to play an important role in testicular development and gametogenesis. However, the structure, expression and function of *PRAMEY* in sheep remain unknown. In this paper, we found that *PRAMEY* was extremely expressed in the testis and significantly upregulated during testicular growth. The mRNA abundance of *PRAMEY* was significantly correlated with testicular size traits in Hu sheep at 6 months. These results imply that *PRAMEY* may play an important role in testicular growth.

**Abstract:**

*PRAMEY* (preferentially expressed antigen in melanoma, Y-linked) belongs to the cancer-testis antigens (CTAs) gene family and is predominantly expressed in testis, playing important roles in spermatogenesis and testicular development. This study cloned the full-length cDNA sequence of ovine *PRAMEY* using the rapid amplification of cDNA ends (RACE) method and analyzed the expression profile and copy number variation (CNV) of *PRAMEY* using quantitative real-time PCR (qPCR). The results revealed that the *PRAMEY* cDNA was 2099 bp in length with an open reading frame (ORF) of 1536 bp encoding 511 amino acids. *PRAMEY* was predominantly expressed in the testis and significantly upregulated during postnatal testicular development. The median copy number (MCN) of *PRAMEY* was 4, varying from 2 to 25 in 710 rams across eight sheep breeds. There was no significant correlation between the CNV of *PRAMEY* and testicular size, while a significant positive correlation was observed between the mRNA expression and testicular size in Hu sheep. The current study suggests that the expression levels of *PRAMEY* were closely associated with testicular size, indicating that *PRAMEY* may play an important role in testicular growth.

## 1. Introduction

Sheep (*Ovis aries*) is one of the most economically important domesticated livestock providing necessary resources for humans, such as meat, milk, wool, etc. The male reproductive capacity is paramount to improve the profitability of the sheep industry [1]. Identifying genetic markers for the early evaluation of ram fertility can accelerate the genetic improvement of male reproduction. To date, very limited genetic markers have been identified for male fertility in sheep [2,3]. The Y-chromosome is present in male mammals only and contains a cluster of testis-specific genes essential for spermatogenesis and male fertility, the characterization of which can provide important clues that can deepen our understanding of the underlying mechanism of male infertility and subfertility [4,5]. Recent progress on delineating the structure and gene function of the Y-chromosome in humans [6], chimpanzees [7], rhesus macaques [8], mice [9], and bovines [10] has identified many protein-coding gene families involved in spermatogenesis and male fertility. The annotated assembly of the ovine Y-chromosome has been hampered by the highly repetitive DNA sequence [11,12], and little is known about the function of Y-linked protein-coding gene families in sheep.

Comparative analyses of mammalian Y-chromosomes have revealed that largely amplified gene families dominate the ampliconic region of the male-specific region of the mammalian Y-chromosome (MSY) [6,10,13]. Expression analyses of multicopy genes in the MSY have found that they are expressed predominantly or exclusively in testis, which suggests they are essential for spermatogenesis and male fertility [4,14]. In addition, the copy number variation (CNV) of these Y-linked multicopy genes was significantly associated with male reproduction in humans [15,16] and cattle [17,18,19]. In sheep, at least four multicopy gene families have been identified in the draft assembly of the ovine MSY [12], including *HSFY* (heat-shock transcription factor, Y-linked), *ZNF280BY* (zinc finger protein 280B-like, Y-linked), *ZNF280AY* (zinc finger protein 280A-like, Y-linked), and *PRAMEY* (preferentially expressed antigen in melanoma, Y-linked).

*PRAMEY* is a member of multicopy gene families identified only in the bovid lineages, which belongs to a family of cancer-testis antigens (CTAs) that are expressed predominantly in testis and a wide variety of tumors with functions in spermatogenesis and immunity [20,21]. Phylogenetic analysis indicated that bovine *PRAMEY* originated through the autosome-to-Y transposition of a gene block comprising *ZNF280B*/*ZNF280A*/*PRAME* on bovine chromosome 17 and was separately amplified to 10 copies on the bovine Y-chromosome [10,21]. The mouse *Pramel1* (Prame-like 1), an ortholog of bovid *PRAME*/*PRAMEY*, is expressed solely in testis, and its mRNA and protein expression levels are continually increasing during testes development, suggesting that *PRAMEY* is required for male fertility [22]. In cattle, *PRAMEY* protein was found to be predominantly expressed in testis and migrated with the expansion of the acrosome granule in the round spermatids during acrosomal biogenesis, suggesting an essential role in normal acrosome formation during spermiogenesis [23]. Additionally, a previous study showed that the CNV of *PRAMEY* is negatively associated with bull fertility, confirming its role in regulating male fertility [17]. Therefore, clarification of the expression profile and CNV of ovine *PRAMEY* will provide an important clue to identifying a novel candidate gene for the early selection of elite sires in sheep-breeding programs.

To date, no study has been conducted to characterize the molecular structure and function of the ovine *PRAMEY* [24]. This study aimed to clone the full-length cDNA of ovine *PRAMEY* and determine its expression patterns, as well as to investigate its CNV in different sheep breeds and the association of CNV with testicular size in Hu sheep. To our knowledge, this study is the first to clone the full-length cDNA sequence of ovine *PRAMEY* and find that *PRAMEY* was highly expressed in testis and that its mRNA expression was highly correlated with testicular growth, which provides a clue for functional analysis in the future.

## 2. Materials and Methods

### 2.1. Sample and Testicular Traits Collection

The samples were divided into three groups based on the purpose of this study (Table 1). Group I contained 137 blood samples from six introduced sheep breeds (Suffolk, *n* = 26; East Friesian, *n* = 6; South African Mutton Merino, *n* = 17; Texel, *n* = 27; Dorper, *n* = 20; White Suffolk, *n* = 29) and one Chinese indigenous sheep breed (Tan sheep, *n* = 12). These individuals were yearling rams without any phenotypic records, which were used to estimate the CNV of *PRAMEY* among different sheep breeds. Group II contained 573 Hu sheep, which are recognized for their precocious puberty and high prolificacy. These individuals were weaned at 2 months and reared in single pens under the same dietary and management conditions at the Minqin Defu Argi-Tech Co., Ltd. (Longitude: E103°08′; Latitude: N38°62′; Minqin, China). They were slaughtered to collect testis tissue and testicular size records at 6 months, including left and right testicular weight (LTW, RTW), left and right testicular volume (LTV, RTV), left and right testicle length (LTL, RTL), left and right testicle width (LTWI, RTWI), left and right epididymal weight (LEW, REW), scrotal circumference (SC), total testicular weight (TTW), total testicular volume (TTV), total epididymal weight (TEW), and testicular index (TI, the ratio of total testis weight to body weight before slaughter). Meanwhile, 100 out of the 573 rams at 6 mo were randomly selected to test the relationship of *PRAMEY* mRNA expression with its CNV and testis size. Moreover, 6 out of the 573 Hu sheep with similar body weight at 6 mo were selected to collect tissue samples of heart, liver, spleen, lung, kidney, and muscle for expression pattern analysis. Group III contained testis samples from three healthy male Hu sheep slaughtered at 0 d (infant), 3 mo (puberty), 6 mo (sexual maturity), and 12 mo (body maturity), respectively, which were used to investigate the *PRAMEY* expression profile during testicular development. In addition, ovary tissues were collected from three adult female Hu sheep for negative controls. All tissue samples were collected in 2 mL Eppendorf tubes, rapidly frozen in liquid nitrogen, and stored at −80 °C for subsequent DNA and RNA extraction.

### 2.2. DNA, Total RNA Extraction and cDNA Synthesis

Genomic DNA was extracted from 500 μL of blood and 50–100 mg of testis and ovary samples using a standard phenol-chloroform method [25]. Total RNA was isolated from 50–100 mg of tissue samples (heart, liver, spleen, lung, kidney, muscle, testis, and ovary) using the RNAsimple Total RNA Kit (Tiangen Biotech, Beijing, China) and reverse-transcribed with PrimeScript^TM^ RT Master Mix (Takara Bio, Shiga, Japan) according to the manufacturer’s instructions. The concentrations of DNA and RNA were quantified by a NanoDrop 2000 spectrophotometer (Thermo Fisher Scientific, Waltham, MA, USA), and the integrity was assessed by 1% agarose gel electrophoresis; the samples were then stored at −80 °C for further analysis.

### 2.3. Full-Length cDNA Cloning of Ovine PRAMEY

The primers for the middle fragment of the *PRAMEY* cDNA sequence were designed using Premier v. 6.0 software (Premier Biosoft Interpairs, Palo Alto, CA, USA) based on the bovine *PRAMEY* cDNA sequence (GenBank acc. no. GU144301). The middle fragment of *PRAMEY* cDNA was amplified from testis cDNA by reverse transcription polymerase chain reaction (RT-PCR) with *PRAMEY*1 primers (Table 2). RT-PCR amplifications were performed in a volume of 25 µL containing 1 μL of cDNA (50 ng/μL) or distilled water, 0.5 μL of each primer (10 μmol/μL), 10.5 μL of distilled water, and 12.5 μL of 2× Easy Taq PCR Super MIX (TransGen Biotech). The amplification conditions consisted of 95 °C for 5 min, followed by 35 cycles of denaturation at 94 °C for 30 s, annealing at 62 °C for 30 s, extension at 72 °C for 30 s, and a final extension at 72 °C for 5 min. The amplification products were electrophoretically separated on 1.5% agarose gel, purified with the AxyPrep DNA Gel Extraction Kit (Axygen Scientific Inc., Union City, CA, USA), cloned using a Zero Background pTOPO-Blunt Simple Cloning Kit (Aidlab Biotechnologies Co., Ltd., Beijing, China), and transformed into EPI300 competent cells. The confirmed clones were selected and sequenced by Sangon Biotech Co., Ltd., Shanghai, China. Based on the obtained cDNA fragment of ovine *PRAMEY*, gene-specific primers (GSPs, Table 2) were designed to amplify the 3′ and 5′ ends of *PRAMEY* cDNA by rapid amplification of cDNA ends (RACE) using the SMART RACE cDNA amplification kit (Takara, Shiga, Japan) following the manufacturer’s protocol.

### 2.4. Bioinformatic Analysis

The open reading frame (ORF) and amino acid sequence of ovine *PRAMEY* cDNA were deduced using the ORF Finder (http://www.ncbi.nlm.nihgov/gorf/orfig.cgi, accessed on 15 January 2021) and the DNAMAN software (Lynnon Corporation, Pointe-Claire, QC, Canada), respectively. The full-length cDNA sequence of ovine *PRAMEY* identified in our study was aligned to the ovine Y-chromosome draft sequence assembly (GenBank accession no. CM022046) using Splign (https://www.ncbi.nlm.nih.gov/sutils/splign/splign.cgi?textpage=online&level=form, accessed on 15 January 2021). Protein molecular weight and theoretical isoelectric point (pI) were calculated via the ExPASy Compute pI/Mw tool (http://web.expasy.org/compute_pi/, accessed on 15 January 2021). Protein domains were predicted by SMART (http://smart.embl-heidelberg.de/, accessed on 15 January 2021). Transmembrane domains were predicted by the TMHMM 2.0 software (http://www.cbs.dtu.dk/services/TMHMM-2.0, accessed on 15 January 2021). Putative signal peptides sequence was identified using SignalP software (http://www.cbs.dtu.dk/services/SignalP-4.0, accessed on 15 January 2021). Secondary and tertiary structures were predicted by SOPMA (http://npsa-pbil.ibcp.fr/, accessed on 15 January 2021) and Swiss-model programs (http://swissmodel.expasy.org, accessed on 15 January 2021), respectively. Potential phosphorylation and glycosylation sites were predicted by NetPhos (http://www.cbs.dtu.dk/services/NetPhos/, accessed on 15 January 2021) and Net OGlyc (http://www.cbs.dtu.dk/services/NetOGlyc/, accessed on 15 January 2021) programs, respectively. The homology search for code sequences (CDS) was obtained by the BLAST server in the NCBI program (https://blast.ncbi.nlm.nih.gov/Blast.cgi, accessed on 15 January 2021). The phylogenetic tree of *PRAMEY*/*PRAME* proteins was generated using the neighbor-joining method and constructed by MEGA 7.0 software [26].

### 2.5. Estimation of mRNA Expression Levels and CNV of Ovine PRAMEY

The primers for mRNA expression were designed based on ovine *PRAMEY* cDNA sequences obtained from full-length cDNA cloning in our study (Table 2). The expression pattern of *PRAMEY* in different tissues (heart, liver, spleen, lung, kidney, muscle, testis, and ovary) was preliminarily analyzed by semi-quantitative RT-PCR (sqRT-PCR) assay using *β-actin* as an internal control. The amplification reaction was performed with the same components and amplification conditions as those used to amplify the middle fragment of *PRAMEY* cDNA described above, except using the corresponding primers and annealing temperature listed in Table 2. Subsequently, the mRNA expression levels of ovine *PRAMEY* in four stages during testis postnatal development, seven tissues, and large- and small-testis groups of 6 mo Hu sheep were further investigated by quantitative real-time PCR (qPCR). The quantitative calculation was performed using the 2^−ΔΔCt^ method with *β-actin* as the reference gene. The qPCR experiment was performed using 1 μL of cDNA (5 ng/μL), 5 μL of SYBR Premix Ex Taq II (TaKaRa, Dalian, China), 0.5 μL of each primer (10 μmol/μL), and 3 μL of distilled H_2_O. Amplification reactions were 94 °C for 5 min followed by 34 cycles of 94 °C for 30 s and 58 °C for 30 s, finishing with elongation at 72 °C for 30 s. The melting curves were obtained by increasing the temperature from 65 °C to 95 °C with an increment of 0.5 °C for 5 s. Every sample was run in triplicate.

To measure the copy number (CN) of *PRAMEY* across 710 rams from 8 sheep breeds, qPCR was performed using the SYBR Premix Ex TaqII Kit (TaKaRa, Dalian, China) with the CFX96 real-time PCR detection system (Bio-Rad, Hercules, CA, USA). A single copy gene *DDX3Y* (DEAD box polypeptide 3, Y-linked, GenBank acc. No. NT182066) was used as a reference [12]. Standard curves were generated from a ram genomic DNA diluted to 50, 25, 12.5, 6.25, and 3.125 ng/μL for both *PRAMEY* and *DDX3Y* primers (Table 2). The resulting reactions had a primer efficiency of 2.04 and 2.09 for *PRAMEY* and *DDX3Y*, respectively, according to the equation E = 10^1/slope^ [17,27]. Within each 96-well PCR plate, one fixed ram sample was used as a calibrator to minimize inter-assay variation, and one female genomic DNA sample and one distilled water sample were used as a negative and blank control, respectively. All samples were assayed in triplicate. qPCR was performed in a 10 µL reaction with 5 µL of SYBR, 0.5 µL of each primer (10 μmol/μL), 1 µL DNA (5 ng/µL), and 3 µL of distilled H_2_O. The amplification program was 95 °C for 3 min; 39× (95 °C for 10 s, 63 °C for 30 s, and 72 °C for 30 s), then 72 °C for 5 min. Melting curves were generated every 0.1 °C from 60–95 °C. Subsequently, the CN of *PRAMEY* was determined using a method described previously [19].

### 2.6. Association and Statistical Analysis

All statistical analyses were performed using SPSS 25.0 software (SPSS, Inc, Chicago, IL, USA). To minimize the technical error, the raw qPCR data of each sample that showed a coefficient of variation (CV) < 1% among triplicates were used for further analysis. The normal distribution of the *PRAMEY* CN data was tested using the Kolmogorov–Smirnov and Shapiro–Wilk normality tests [28,29]. Boxplot analyses were conducted with the CN data to identify outliers [30]. The nonparametric Mann–Whitney U test was applied to compare the median copy number (MCN) of *PRAMEY* between sheep breeds [31], and *p*-values were then corrected for multiple testing using Bonferroni correction [32]. Levene’s test was used to test the homogeneity of group variances. Differences in mRNA expression levels of *PRAMEY* among seven tissues and different ages of testis were analyzed using one-way ANOVA and LSD post hoc test. Welch’s t-test was conducted to compare the difference in mRNA expression levels between the large- and small-testis groups (unequal variances). Correlation analyses between CNV and testicular size, CNV and mRNA expression level, and mRNA expression level and testicular size were performed for Hu sheep using Spearman’s correlation test. A *p*-value of ≤ 0.05 was regarded as statistically significant for all tests.

## 3. Results

### 3.1. Molecular Characteristics of Ovine PRAMEY cDNA

The full-length cDNA sequence (GenBank acc. no. MN850838) was 2099 bp, containing 233 bp of 5′ untranslated region (5′-UTR), a 1535 bp ORF encoding a polypeptide of 511 amino acids, and a 330 bp 3′-UTR including a poly A signal sequence (Appendix A). Based on alignment with CM022046, we found that the genomic sequence of ovine *PRAMEY* was 5687 bp, including five exons and four introns (Figure 1). Bioinformatic analysis revealed that the predicted molecular weight of *PRAMEY* was 57.95 kDa, with a theoretical isoelectric point (pI) of 6.57. Glu (19.4%), Asp (7.8%), and Asn (6.7%) were the most abundant amino acids in the putative *PRAMEY*. Moreover, the instability, grand average hydrophilicity (GRAVY), and aliphatic index for PRAMEY were 51.78, 0.041, and 108.4, respectively, suggesting it was unstable, hydrophobic, and thermally stable. *PRAMEY* is a non-secreted protein without signal peptide and does not contain a transmembrane domain (Appendix A). Moreover, *PRAMEY* protein was predicted to contain 2 putative N-glycosylation sites and 43 putative phosphorylation sites (31 Ser, 10 Thr, and 2 Tyr) (Appendix A). Secondary structure prediction showed that *PRAMEY* was mainly composed of α-helix (55.19%), extended strand (36.59%), and random coil (8.22%) (Appendix A). The tertiary structure of *PRAMEY* confirmed a typical horseshoe-like shape structure (Appendix A).

### 3.2. Homology and Phylogenetic Analysis

Multiple sequence alignment showed that ovine *PRAMEY* cDNA shared 98% sequence identity with goat *PRAME* (XM_018045004), 89% identify with bovine *PRAMEY/PRAME* (GU144302 and NM_001245953), and 84% identify with its paralogs on the ovine chromosome 17 (XM_027980272). The deduced amino acid sequences of the retrieved *PRAMEY*/*PRAME* orthologous sequences from 11 mammalian species were used to construct a phylogenetic tree using the neighbor-joining (NJ) method. The result showed that ovine *PRAMEY* and bovine *PRAMEY* were clustered into a subclade, and together with bovine *PRAME* on chromosome 17 formed one clade. The autosomal orthologues of ovine *PRAMEY* in eight mammalian species were clustered together with high bootstrap values (Figure 2).

### 3.3. Expression Profile of the Ovine PRAMEY

Before the qPCR analysis, a conventional PCR was run to validate the male specificity of the *PRAMEY* and *DDX3Y* primers. The results revealed that all primers used only have amplification in male genomic DNA or testis cDNA (Figure 3 and Figure 4), indicating that the designed primers were male-specific and can be used for the subsequent analysis.

The expression patterns of ovine *PRAMEY* were investigated by sqRT-PCR and qPCR. sqRT-PCR analysis across seven tissues revealed that *PRAMEY* was expressed only in the testis (Figure 4). In contrast, qPCR results showed that *PRAMEY* had a very weak expression in muscle when compared to testis expression (*p* < 0.001; Figure 5A). Moreover, qPCR analysis revealed that the expression of *PRAMEY* was gradually upregulated during testicular development (0 d, 3, 6, and 12 mo of age). *PRAMEY* had a low expression in newborn and 3-month-old testes but increased significantly in 6- and 12-month-old testes (*p* < 0.05; Figure 5B).

### 3.4. CNV of PRAMEY across Sheep Breeds

The CN of *PRAMEY* was 4 for the calibrator sample using the equations for CN calculation. Meanwhile, the CN showed significant variation between and within the sheep breeds investigated. The MCN of 710 rams from eight sheep breeds was 4, ranging from 2 to 25 copies. Of those, Tan sheep showed the lowest MCN (3 copies, range: 2–7), and East Friesian sheep possessed the highest MCN (7 copies, range: 4–9) (Table 1). The CN data for all sheep breeds did not fit normal distribution (*p* < 0.05) based on the Kolmogorov–Smirnov and Shapiro–Wilk normality tests. One and eight outliers in White Suffolk and Hu sheep were identified by the boxplot procedure, respectively (Appendix A). Moreover, the pairwise comparisons of the MCN revealed significant differences between sheep breeds. In general, both Dorper and White Suffolk sheep had significantly higher *PRAMEY* MCN than Hu and Tan sheep (*p* < 0.05; Appendix A).

### 3.5. The Relationship among PRAMEY CNV, mRNA Expression in Testis, and Testicular Size

Within 573 rams with CN, the mRNA expression level in the testis of 100 rams was investigated using qPCR. Spearman’s correlation analysis revealed that no significant relationship was found between *PREMEY* CNV and mRNA expression levels (r = 0.043, *p* = 0.682). After excluding eight outliers, association analysis revealed that the CNV of *PRAMEY* had no significant relationship with any testicular traits in Hu sheep (*p* > 0.05; Table 3). Contrastively, the mRNA expression level of *PRAMEY* had a positive correlation with testicular traits in 100 rams (*p* < 0.01; Table 3; Appendix A).

## 4. Discussion

In this study, we firstly cloned the full-length cDNA of ovine *PRAMEY* and estimated its MCN was 4, and the CN varied from 2 to 25 in 710 rams across eight sheep breeds. The *PRAMEY* was specifically expressed in testis and significantly upregulated during testicular development. Association analysis revealed that the CNV of *PRAMEY* had no significant relationship with testicular size, while its mRNA expression levels in the testis were positively associated with testicular size in Hu sheep. Our study suggested that *PRAMEY* plays an important role in postnatal testicular growth in Hu sheep.

To date, *PRAMEY* is known to be amplified only in bovid species during evolution, with approximately 10 copies on the bovine Y-chromosome [10]. Our study found that the MCN of *PRAMEY* was 4 copies and varied from 2 to 25 in 710 rams across eight sheep breeds, which indicated that *PRAMEY* has restricted amplification on the ovine Y-chromosome. As mentioned previously, the sequence structure and gene content of the mammalian Y-chromosomes appear to have extraordinary divergences among the different species [4,7,8]. Additionally, it is notable that the cattle (*Bos taurus*) Y-chromosome is submetacentric, whereas the sheep (*Ovis aries*) Y-chromosome is small metacentric [33]. Therefore, we extrapolated that the remarkable difference in the Y-chromosome among the bovid lineages may account for the difference in the *PRAMEY* CN between the bovine and ovine.

Previous studies have reported that bovine *PRAMEY* has testis-specific expression and is upregulated during testicular growth. Moreover, the sense RNA of *PRAMEY* is detected only in spermatids, whereas the antisense RNA is detected across all cells of seminiferous tubules and with the highest expression in spermatids [21]. Furthermore, *PRAMEY* protein was found to be highly enriched in migrated acrosomal granule cells during acrosomal biogenesis [23]. In our study, *PRAMEY* showed similar expression patterns in ovine testis tissues, and the mRNA expression of *PRAMEY* was significantly greater in the large-testes group than in the small-testes group. Histological analysis has shown that large testes possessed a longer diameter of seminiferous tubules and a greater number of mature sperm than small testes [34]. Our study implies that *PRAMEY* may play a key role in testicular growth and spermatogenesis [12,23]. However, further determination of the precise cellular and subcellular localization of *PRAMEY* protein in germ cells would provide useful insights into the functions of the ovine *PRAMEY* in spermatogenesis and testicular development.

The amplification and maintenance of Y-linked testis-specific genes in mammals play crucial roles in testis development and spermatogenesis [35]. For instance, the bovine *PRAMEY* and its two orthologues, mouse *Pramel1* (Prame-like 1) [22] and human *PRAME* [36], were localized exclusively to the acrosome of mature spermatozoa, indicating that they are involved in acrosome formation, which is a crucial process in the last stages of spermiogenesis for sperm maturation [37]. The results from our study found that the mRNA expression of *PRAMEY* was significantly positively associated with testis size in Hu sheep at 6 months. It seems reasonable to speculate that the more abundant expression of *PRAMEY* in the large testis was required to produce more spermatozoon containing a well-developed acrosome.

It was proposed that the biological roles of the Y-linked gene families were closely related to testicular development and spermatogenesis because they were specifically expressed in testis and massively amplificated on the Y-chromosome [38,39]. Numerous studies have confirmed that the CNVs of Y-linked gene families were significantly associated with testis size and semen quality [19]. For example, the CNV of *PRAMEY* was negatively correlated with scrotal circumference (SC), relative scrotal circumference (RLSC), percentage of normal sperm (PNS), and non-return rate (NRR) in Holstein bulls, indicating that the low CN of *PRAMEY* was beneficial for bull fertility [17]. However, our study found no significant correlation between the CNV of *PRAMEY* and any testicular traits in Hu sheep, suggesting *PRAMEY* CNV maybe represent a benign variation and does not cause phenotypic differences in Hu sheep [40,41].

As mentioned in a previous review [40], CNV can contribute to phenotypes via molecular mechanisms such as gene dosage effects, position effects, and gene disruption. For example, the deletion of the CN of *RBMY1* (RNA binding motif protein Y-linked family 1) led to insufficient dosage of the mRNA and protein levels resulting in asthenozoospermia and sterility in men [15]. Therefore, we analyzed the correlation between the CNV of *PRAMEY* and its mRNA in testis, and no significant association was observed, which suggested that the ovine *PRAMEY* may have not a dosage effect in testis. Notably, the high degree of sequence similarity between functional copies and pseudogenes may overestimate the functional copy number, as evidenced by previous studies on the CN of *ZNF280AY* and *RBMY1* [15,19]. Until recently, it has been difficult to design primers that capture all of the functional copies of the *PRAMEY* gene because the ovine Y-chromosome is not well-annotated. Therefore, further structural description of *PRAMEY* based on a comprehensive reference sequence of ovine Y-chromosome will facilitate accurate detection of functional copies and determine the relationship between *PRAMEY* CN and its mRNA and protein levels in testis.

## 5. Conclusions

Our results provide evidence that *PRAMEY* is a testis-specific gene family on the ovine Y-chromosome and shows CNV between and within sheep breeds. The mRNA expression level of *PRAMEY* is significantly associated with testicular size, suggesting that *PRAMEY* may play an essential role in testicular growth and spermatogenesis.

## Figures and Tables

**Figure 1 animals-12-02380-f001:**
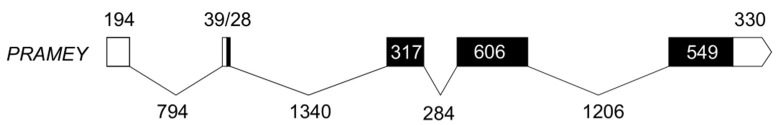
Genome structures of the ovine *PRAMEY*. *PRAMEY* contains five exons and four introns. Black boxes represent coding regions, and white boxes represent 5′ UTR and 3′ UTR regions. The UTR, introns, and exons are drawn to scale.

**Figure 2 animals-12-02380-f002:**
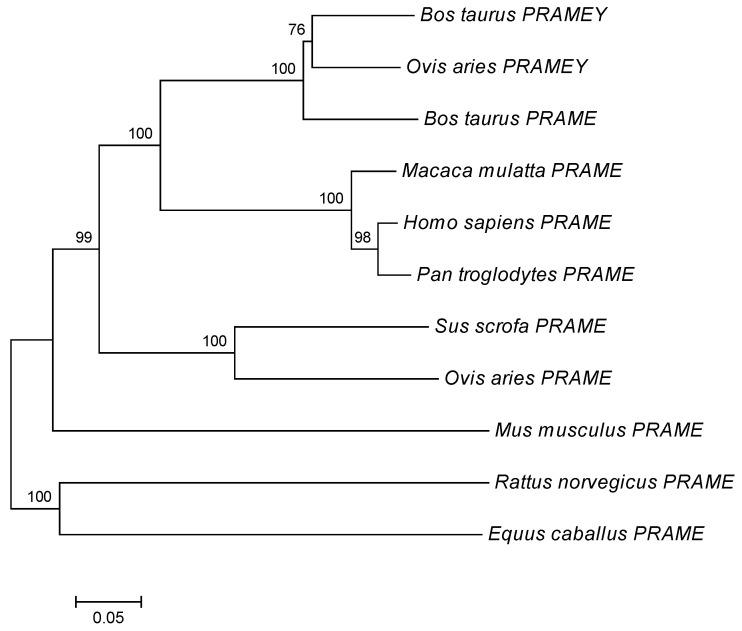
Neighbor-joining tree of *PRAMEY*/*PRAME* proteins. The GenBank accession numbers of *PRAMEY* protein are *Bos taurus*, ADP21954, and *PRAME* protein are *Bos taurus*, NP_001232882; *Macaca mulatta*, AFJ71405; *Homo sapiens*, CAG30435; *Pan troglodytes*, NP_001267447; *Sus scrofa*, XP_020953254; *Ovis aries*, XP_027831889; *Mus musculus*, NP_001108549; *Rattus norvegicus*, XP_001055257.2; *Equus caballus*, XP_023502450. The numbers on the branches represent 1000 bootstrap values.

**Figure 3 animals-12-02380-f003:**
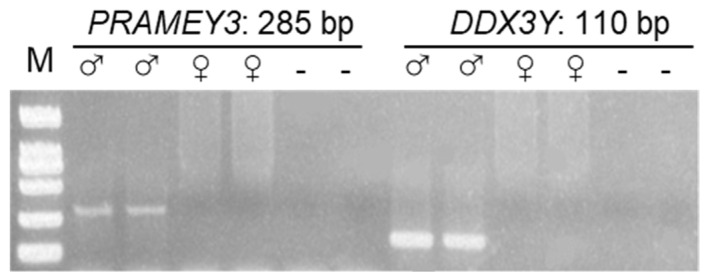
Gel electrophoresis of conventional PCR products of ovine *PRAMEY3* and *DDX3Y*. The primers of these two Y-linked genes amplified male-specific bands with expected fragment sizes labeled above each band. M: 2 kb DNA ladder; ♂: male sheep genomic DNA; ♀: female sheep genomic DNA; −: negative control (distilled water).

**Figure 4 animals-12-02380-f004:**
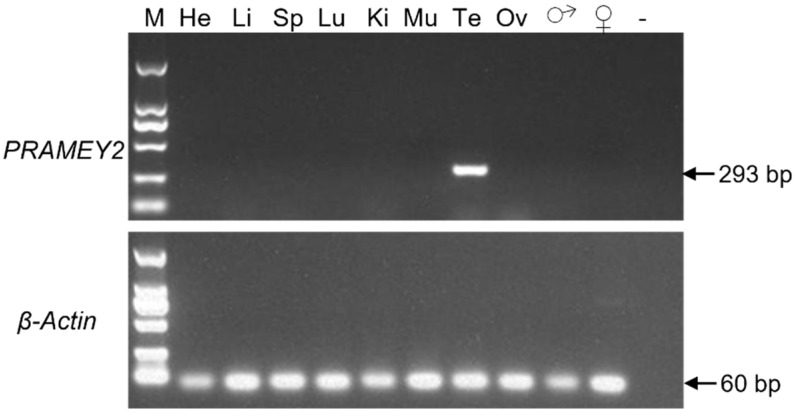
Tissue expression pattern of ovine *PRAMEY*. The *PRAMEY* was expressed specifically in testis. The *β-Actin* was used as an internal control for semi-quantitative RT-PCR. M: 2 kb DNA ladder; He: heart; Li: liver; Sp: spleen; Lu: lung; Ki: kidney; Mu: muscle; Te: testis; Ov: ovary; ♂: male sheep genomic DNA; ♀: female sheep genomic DNA; −: negative control (distilled water).

**Figure 5 animals-12-02380-f005:**
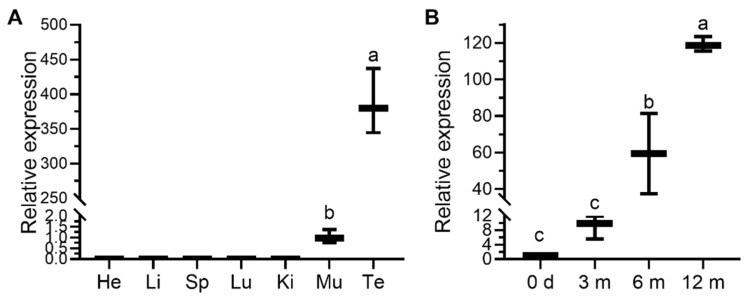
Relative expression levels of the *PRAMEY* mRNA in Hu sheep were detected by qPCR. (**A**) The expression level of *PRAMEY* in seven different tissues at 6 months old. *PRAMEY* was expressed mainly in testis and slightly in muscle. He: heart; Li: liver; Sp: spleen; Lu: lung; Ki: kidney; Mu: muscle; Te: testis. (**B**) mRNA expression abundance of *PRAMEY* at different developmental stages of testis. The expression of *PRAMEY* mRNA was expressed very low at 0 days and 3 months but significantly upregulated at 6 and 12 months. Values are means ± SD of three biological replicates (*n* = 3). Different letters above error bars indicate significant differences, *p* ≤ 0.05.

**Table 1 animals-12-02380-t001:** Sample information and the median copy number of *PRAMEY* in 8 sheep breeds.

Group	Breed (Full Name)	Short Name	Sample Size	Sample Type	Median Copy Number (Range)
Group I	Dorper	DP	20	Blood	6 (4–9)
	East Friesian	EF	6	Blood	7 (4–9)
	Suffolk	SK	26	Blood	5 (3–9)
	South African Mutton Merino	SMM	17	Blood	5 (4–9)
	Tan sheep	TS	12	Blood	3 (2–7)
	Texel	TL	27	Blood	5 (2–7)
	White Suffolk	WSK	29	Blood	6 (2–25)
Group II	Hu sheep	HS	573 ^1^	Testis, (Heart, Liver, Spleen, Lung, Kidney, Muscle) ^2^	4 (2–12)
Group III	Hu sheep	HS	12	Testis	

^1^ A male lamb of Hu sheep was selected randomly as the calibrator to minimize inter-assay variation. ^2^ Tissue samples were collected from 6 individuals with similar body weight from 573 Hu sheep at 6 months.

**Table 2 animals-12-02380-t002:** The information on primers used in the present study.

Primer Name	Primer Sequence (5′-3′)	PCR Product Length (bp)	Annealing Temperature (°C)	Usage
*PRAMEY*1	F:CGGAGTAGGTTCACGATGGG	1112	62	RT-PCR
	R:TCTAGGTCCTGTAGGGTGGC			
5′GSP	GATTACGCCAAGCTTTCTAGGTCCTGTAGGGTGGC	-	-	5′-RACE
3′GSP	GATTACGCCAAGCTTCGCATCTGGGACGGATGGGA	-	-	3′-RACE
*PRAMEY*2	F:TGGAGGTGAACTGCATCTGG	293	58	sqRT-PCR and qPCR ^1^
	R:TGAAAGCAGGCAGTTGGTGA			
*β-actin*	F:CCTGCGGCATTCACGAA	60	60	
	R:GCGGATGTCGACGTCCACA			
*PRAMEY*3	F:TGCCCTCATTTCGTTACCCT	285	60	qPCR ^2^
	R:TTTTCGCTTAGTTATCTGCTATCAT			
*DDX3Y*	F:TCGCCGCTTGCTTACGTACACT	110	69	
	R:ACACCCTCTGGTTAACGGCCAT			

^1^ qPCR was used to estimate the mRNA expression levels of *PRAMEY*. ^2^ qPCR was used to estimate the copy number of *PRAMEY*.

**Table 3 animals-12-02380-t003:** Spearman’s correlation between the copy number and mRNA expression levels of *PRAMEY* and the testicular traits in Hu sheep.

Testicular Trait ^1^	Mean ± SD	*PRAMEY* Copy Number		*PRAMEY* mRNA	
		r	*p*	r	*p*
TI	5.29 × 10^−3^ ± 2.06 × 10^−3^	0.018	0.676	0.602	<0.0001
TTW (g)	238.79 ± 92.67	0.012	0.766	0.609	<0.0001
LTW (g)	119.56 ± 46.75	0.002	0.966	0.608	<0.0001
LTL (mm)	81.05 ± 12.54	−0.056	0.248	0.269	0.008
LTWI (mm)	57.73 ± 8.96	−0.050	0.305	0.323	0.001
LEW (g)	18.15 ± 4.45	−0.013	0.750	0.551	<0.0001
RTW (g)	119.23 ± 46.64	0.023	0.583	0.621	<0.0001
RTL (mm)	80.56 ± 12.53	−0.017	0.731	0.299	0.003
RTWI (mm)	57.78 ± 8.90	−0.027	0.577	0.325	0.001
REW (g)	17.95 ± 4.44	0.018	0.674	0.532	<0.0001
TEW (g)	36.05 ± 8.64	0.007	0.865	0.499	<0.0001
LTV (mL)	104.97 ± 45.70	0.056	0.184	-	-
RTV (mL)	103.56 ± 43.77	0.056	0.178	-	-
TTV (mL)	208.53 ± 88.65	0.056	0.180	-	-
SC	23.13 ± 3.34	0.029	0.495	-	-

^1^ TI = testicular index; TTW = total testicular weight; LTW = left testicular weight; LTL = left testicle length; LTWI = left testicle width; LEW = left epididymal weight; RTW = right testicular weight; RTL = right testicle length; RTWI = right testicle width; REW = right epididymal weight; TEW = total epididymal weight; LTV = left testicular volume; RTV = right testicular volume; TTV = total testicular volume; SC = scrotal circumference.

## Data Availability

The data presented in this study are available on request from the corresponding author.

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
