# Peer review of "PRAMEY: A Bovid-Specific Y-Chromosome Multicopy Gene Is Highly Related to Postnatal Testicular Growth in Hu Sheep"

_animals, 2022, doi:10.3390/ani12182380_

Round 1

Reviewer 1 Report

The authors describe a multicopy gene located on Y chromosome in sheep and they have found a significant positive correlation between mRNA level of PRAMEY and testis size.

The laboratory techniques and methodology were appropriate. Negative controls were properly used.

I have noticed one thing to remedy. The authors describe bioinformatic analyses (section 2.4.) including how they determined isoelectric point, different domains, sequences, and structures. Most of them are not presented in the manuscript. Please provide a picture, where these results are highlighted.

Author Response

Reviewer: 1

The authors describe a multicopy gene located on Y chromosome in sheep and they have found a significant positive correlation between mRNA level of PRAMEY and testis size.

The laboratory techniques and methodology were appropriate. Negative controls were properly used.

I have noticed one thing to remedy. The authors describe bioinformatic analyses (section 2.4.) including how they determined isoelectric point, different domains, sequences, and structures. Most of them are not presented in the manuscript. Please provide a picture, where these results are highlighted.

Response: We highly appreciate your kind and valuable suggestion. Based on the bioinformatic analysis of the predicted PRAMEY protein, the figures have been integrated into a picture and presented in supplementary figure 2, and described it in result part. Please check the supplementary figure 2 and Lines 239-246 in the revised manuscript.

Reviewer 2 Report

Dear Authors,

I have read with interest your article titled "PRAMEY: a bovid specific Y-chromosome multicopy gene is highly related to testicular development in Hu sheep", submitted for publication in Animals. My general opinion is good, and I believe that what has been presented deserves publication. However, I think it is important that you consider some suggestions, listed below, which in my opinion can improve both the understanding of the results obtained and better clarify the methodology you have followed.

General Comments

1. By looking in the database (NCBI nucleotide) for PRAMEY bovine sequences, 3 results are obtained:

GU144301, GU144302 and NM_001077979. The first 2 belong to a single publication (PLoS ONE 6-2, E16867,2011) and the last to another (Reproduction 153-6, 847-863, 2017). Why in your approach you designed the primers to identify the PRAMEY sequence in the sheep starting from the GU144301 sequence?

2. Why did you not verify the genomic structure of the sequence identified by you (exons / introns)? Yet today there is at least one assembly for the Y chromosome (this also corresponds to the Hu sheep; CM_022046)

3. Considering the whole paper, reference is often made to the fact that the increased expression of PRAMEY in the testis is an indication of importance in the development of the testis. I do not agree: a. I think it is more appropriate to talk about testicle growth, as testicular development is a process that occurs in the fetal period (Payenet al., 1996. The International journal of developmental biology, 40(3), 567–575.). b. The fact that a gene exhibits increased expression as an anatomical structure increases does not mean that this factor is important for this development. The hypothesis that the expression increases because the amount of tissue increases is equally valid. Unfortunately in these cases only a KO experiment can help answer this question.

4. Were there spermatids and spermatozoa in the testicles you took? If so, how can you differentiate expression from tissue or spermatids / spermatozoa?

Others comments

Line 41: I don't think reference number 2 is appropriate when referring to “the early avaluation of ram fertility”. The reference to a semen analysis implies the achievement of sexual maturity.

Line 70: As already mentioned in point 3, I disagree. Also in the reproduction paper (Reproduction 2017 153 847–863) reference is made to “suggest that PRAMEY may play an essential role in acrosome biogenesis and spermatogenesis”.

Line 113: Having 573 subjects and choosing only 3 + 3 to create two groups makes absolutely no sense. I don't think valid results can be obtained from just three subjects. Since I believe you have all the data for all subjects, I suggest recalculating the values considering the 30 subjects with the largest testicles and the 30 subjects with the smallest testicles (about 5% of the total). If this is not possible, then I suggest deleting this result.

Line 148: First it clones and then transforms and not the reverse.

Line 155: I have tried to locate the PRAMEY1 primers on the GU144301 sequence but without success. Can you indicate where they are located on the sequence?

Line 188: If I understand correctly, to measure the expression of PRAMEY 1 you used a single microliter but without indicating the number of replicates. Furthermore this measurement is done in different wells (for PRAMEY and beta-actin) using SYBR. If this is true, then it is a very dangerous procedure and far from what is suggested in the common protocols. 1 uL is a difficult quantity to measure and even small variations can give different results. At least consider 3 or 5 replication.

Line 231: The weight of your protein corresponds to what is expected for bovine PRAMEY protein (reproduction paper) but not its isoform: it would be important to look for this isoform as well. I do not ask for further experiments, but at least you justify this choice to search this form only.

Line 264 (Figure 2): you must indicate PRAMEY3 to immediately understand which primers are being referred to.

Line 269 (Figure 3): you must indicate PRAMEY2 to immediately understand which primers are being referred to. Moreover, Why is there no amplification of the male gDNA samples (are primer locate in different exons)?

Line 280: I believe that the results obtained on 3 subjects are not scientifically relevant.

Line 285: graphs A, B and C must be changed into box-plot form (if 3 + 3 subjects remain, the B must be eliminated)

Line 303: The Y-axis of the supplementary graph S2 needs to be revised. Fix the maximum value at 15

Line 316: a. Indicate the units of measurement; b. Is there any difference in the development of the right testicle compared to the left? C. I suggest presenting at least one X-Y regression chart for a significant value (TTW for example) and all these graphs in supplementary information.

Line 329: As mentioned above, I disagree

Line 342: testicular growth

Line 350: Hypothesis not confirmed by your experiments: a positive correlation does not indicate cause-effect relationship.

Line 358: 3 value are not sufficient.

Author Response

Dear reviewer:

On behalf of authors, I would like to thank you and the two reviewers for the comments and suggestions on our manuscript (animals-1863621). We have studied the valuable comments and revised the manuscript accordingly. The point to point responses to your comments are listed as following:

Reviewer: 2

I have read with interest your article titled "PRAMEY: a bovid specific Y-chromosome multicopy gene is highly related to testicular development in Hu sheep", submitted for publication in Animals. My general opinion is good, and I believe that what has been presented deserves publication. However, I think it is important that you consider some suggestions, listed below, which in my opinion can improve both the understanding of the results obtained and better clarify the methodology you have followed.

General Comments

Comment 1. By looking in the database (NCBI nucleotide) for PRAMEY bovine sequences, 3 results are obtained: GU144301, GU144302 and NM_001077979. The first 2 belong to a single publication (PLoS ONE 6-2, E16867,2011) and the last to another (Reproduction 153-6, 847-863, 2017). Why in your approach you designed the primers to identify the PRAMEY sequence in the sheep starting from the GU144301 sequence?

Response: Thanks for your question. In 2011, Chang et al. firstly identified two bovine PRAMEY transcripts: PRAMEY1 (GenBank acc. no. GU144301) and PRAMEY2 (GenBank acc. no. GU144302). Of which, the length of GU144301 is 2761 bp, while GU144302 is 1899 bp, indicating GU144301 can supply more sequencing information than GU144302 for primer design. In addition, NM_001077979 is the mRNA sequence for its paralogs on the bovine chromosome 17 (PRAME), and sharing 99% sequence identity with GU144302. To amplify male-specific region, this study used GU144301 as a reference sequence to design primers.

Reference: Chang TC, Yang Y, Yasue H, Bharti AK, Retzel EF, Liu WS. The expansion of the PRAME gene family in Eutheria. PLoS One. 2011;6(2):e16867. Published 2011 Feb 10. doi:10.1371/journal.pone.0016867

Comment 2. Why did you not verify the genomic structure of the sequence identified by you (exons / introns)? Yet today there is at least one assembly for the Y chromosome (this also corresponds to the Hu sheep; CM_022046)

Response: Thanks for your suggestion. The full-length cDNA sequence of ovine PRAMEY identified in our study was aligned to the ovine Y chromosome draft sequence assembly (GenBank accession no. CM022046) using Splign (https://www.ncbi.nlm.nih.gov/sutils/splign/splign.cgi?textpage=online&level=form). The genomic sequence of ovine PRAMEY is 5687 bp, including five exons and four introns. The full-length cDNA sequence is 2099 bp, containing 233 bp of 5' untranslated region (5'-UTR), a 1535 bp ORF encoding a polypeptide of 511 amino acids and a 330 bp 3'-UTR including a poly A signal sequence (revision lines 232-233). The gene structure map containing introns has been present in figure 1 in the revised manuscript.

Comment 3. Considering the whole paper, reference is often made to the fact that the increased expression of PRAMEY in the testis is an indication of importance in the development of the testis. I do not agree: a. I think it is more appropriate to talk about testicle growth, as testicular development is a process that occurs in the fetal period (Payenet al., 1996. The International journal of developmental biology, 40(3), 567–575.). b. The fact that a gene exhibits increased expression as an anatomical structure increases does not mean that this factor is important for this development. The hypothesis that the expression increases because the amount of tissue increases is equally valid. Unfortunately in these cases only a KO experiment can help answer this question.

Response: Indeed, you are right and thank you for enlightening us. We realize that what was described is not rigorous. The testis-specific and persistently elevated expression of PRAMEY indicated that PRAMEY mainly functions in postnatal testicular growth and spermatogenesis. To avoid misleading readers, we have made corresponding changes in the whole text.

Comment 4. Were there spermatids and spermatozoa in the testicles you took? If so, how can you differentiate expression from tissue or spermatids / spermatozoa?

Response: Thank you for your question. Hu sheep is one of the most preferable breeds for its precocious puberty and high prolificacy in China, the testis sample (collected at 6 and 12 months) contain both spermatids and spermatozoa. However, we did not differentiate these two cells in our study.

Others comments

Comment 1. Line 41: I don't think reference number 2 is appropriate when referring to “the early evaluation of ram fertility”. The reference to a semen analysis implies the achievement of sexual maturity.

Response: Thank you for your suggestion. Reference number 2 here is indeed inappropriate and has been replaced by one additional reference.

Reference: Notter DR. Genetic aspects of reproduction in sheep. Reprod Domest Anim. 2008;43 Suppl 2:122-128. doi:10.1111/j.1439-0531.2008.01151.x

Comment 2. Line 70: As already mentioned in point 3, I disagree. Also in the reproduction paper (Reproduction 2017 153 847–863) reference is made to “suggest that PRAMEY may play an essential role in acrosome biogenesis and spermatogenesis”.

Response: According to your suggestion, we have re-organized the sentence as “suggesting that PRAMEY is required for male fertility”. Please check Lines 68-69.

Comment 3. Line 113: Having 573 subjects and choosing only 3 + 3 to create two groups makes absolutely no sense. I don't think valid results can be obtained from just three subjects. Since I believe you have all the data for all subjects, I suggest recalculating the values considering the 30 subjects with the largest testicles and the 30 subjects with the smallest testicles (about 5% of the total). If this is not possible, then I suggest deleting this result.

Response: Thanks for your suggestion, we have deleted this result.

Comment 4. Line 148: First it clones and then transforms and not the reverse.

Response: Thank you for your suggestion. We have reviewed to “cloned using a Zero Background pTOPO-Blunt Simple Cloning Kit (Aidlab Biotechnologies Co., Ltd) and transformed into EPI300 competent cells”. Please check Line 144-146 in the revised manuscript.

Comment 5. Line 155: I have tried to locate the PRAMEY1 primers on the GU144301 sequence but without success. Can you indicate where they are located on the sequence?

Response: Thank you for your question. The forward and reverse sequences of PRAMEY1 primers were aligned to the GU144301 sequence via an online tool at https://www.novopro.cn/tools/water.html. The results are listed below.

Comment 6. Line 188: If I understand correctly, to measure the expression of PRAMEY 1 you used a single microliter but without indicating the number of replicates. Furthermore this measurement is done in different wells (for PRAMEY and beta-actin) using SYBR. If this is true, then it is a very dangerous procedure and far from what is suggested in the common protocols. 1 uL is a difficult quantity to measure and even small variations can give different results. At least consider 3 or 5 replication.

Response: We are also very sorry for incorrect description in the common protocols. Indeed, triplicate experiments were performed for the analysis. We added “Every sample was run in triplicate” in revision. Please check line 193 in the revision.

Comment 7. Line 231: The weight of your protein corresponds to what is expected for bovine PRAMEY protein (reproduction paper) but not its isoform: it would be important to look for this isoform as well. I do not ask for further experiments, but at least you justify this choice to search this form only.

Response: Thanks for your suggestion. Liu et al. found that the predicted 58 kDa intact protein was detected in different ages of testis, while the 30 kDa isoform was observed only in testes after puberty and epididymal. This result indicated that PRAMEY plays important role in spermatogenesis and testis development. Given spermatogenesis begins at puberty, we speculated that the 30 kDa isoform acts mainly on spermatogenesis. However, our study aims to investigate the potential function of PRAMEY in postnatal testicular growth. The predicted 57.95 kDa protein of ovine PRAMEY was detected in all four ages of testes. It is thus reasonable to choose this form only.

Comment 8. Line 264 (Figure 2): you must indicate PRAMEY3 to immediately understand which primers are being referred to.

Response: We have made a clear label in Figure 3. Please check Figure 3 in the revision.

Comment 9. Line 269 (Figure 3): you must indicate PRAMEY2 to immediately understand which primers are being referred to. Moreover, Why is there no amplification of the male gDNA samples (are primer locate in different exons)?

Response: We have made the clear label in Figure 4. Please check Figure 4 in the revision. We splined PRAMEY2 to sheep Y chromosome assembly, and found primers spanned an intronic. Therefore, there is no amplification in gDNA.

Comment 10. Line 280: I believe that the results obtained on 3 subjects are not scientifically relevant.

Response: We removed this part following your suggestion.

Comment 11. Line 285: graphs A, B and C must be changed into box-plot form (if 3 + 3 subjects remain, the B must be eliminated)

Response: According to your suggestion. Graphs A and C have been changed into box-plot form and the B has been eliminated. Please check figure 5 in revision.

Comment 12. Line 303: The Y-axis of the supplementary graph S2 needs to be revised. Fix the maximum value at 15

Response: Thank you for the advice. we have fixed the maximum value at 15 on the Y-axis of the supplementary graph S3.

Comment 13. Line 316: a. Indicate the units of measurement; b. Is there any difference in the development of the right testicle compared to the left? C. I suggest presenting at least one X-Y regression chart for a significant value (TTW for example) and all these graphs in supplementary information.

Response: Thank you for the advice, we have marked the units of measurement in Table 3. There is no significant difference in the development of the right testicle compared to the left. In addition, we plotted scatter plots to assess the association between the PRAMEY mRNA expression and testis size (supplementary figure 4).

Comment 14. Line 329: As mentioned above, I disagree

Response: We agree with you and we have rephrased this statement to “Our study suggested that PRAMEY plays an important role in postnatal testicular growth of Hu sheep”. Please check Lines 336-337 in the revision.

Comment 15. Line 342: testicular growth

Response: We have revised “testicular development” to “testicular growth”. Please check Lines 357-358.

Comment 16. Line 350: Hypothesis not confirmed by your experiments: a positive correlation does not indicate cause-effect relationship.

Response: We agree with your opinion. We have reframed this statement to “Our study implies that PRAMEY may play a key role in testicular growth and spermatogenesis”. Please check Lines 367-370 in the revision.

Comment 17. Line 358: 3 value are not sufficient.

Response: Thank you for your suggestion. We have deleted this part in the revised manuscript.

We have tried our best to carefully address all questions/comments/concerns, and sincerely hope that the revised manuscript is acceptable for publication in Animals.

Best wishes!

Yours sincerely

Xiangpeng Yue Ph.D.

Professor of Animal Genetics

College of Pastoral Agriculture Science and Technology,

Lanzhou University,

Lanzhou, Gansu 730020, China.

E-mail: lexp@lzu.edu.cn

Round 2

Reviewer 2 Report

Dear Authors,

I have read your comments and I am satisfied.

I beg you to review this single sentence on line 317: Values are shown as means ± SD. I don't see values shown in the graph.

Best regards

Author Response

Point 1: I beg you to review this single sentence on line 317: Values are shown as means ± SD. I don't see values shown in the graph.

Response 1: Thanks for your suggestion. The figure legend that includes the numbers of replicates and repeats of the techniques used may help to ensure clarity, according to the guidance of writing an effective figure legend (https://www.aje.com/arc/writing-effective-figure-legend/). We revised the sentance into: Values are means ± SD of three biological replicates (n = 3). Please check lines 301-302 in the revised manuscript.
